# Higher Serum E-Selectin Levels Associated with Malignant Brain Edema after Endovascular Thrombectomy for Ischemic Stroke: A Pilot Study

**DOI:** 10.3390/brainsci13071097

**Published:** 2023-07-20

**Authors:** Feng Zhou, Mingyang Du, Yan E, Shuaiyu Chen, Wei Wang, Hongchao Shi, Junshan Zhou, Yingdong Zhang

**Affiliations:** 1Department of Neurology, Nanjing First Hospital, Nanjing Medical University, No. 86 Changle Road, Nanjing 210000, China; snakezf@126.com (F.Z.); joe970708@163.com (Y.E.); 18263707919@163.com (S.C.); dr_wangw@163.com (W.W.); shcao12345678@sina.com (H.S.); zhoujunshan6807@163.com (J.Z.); 2Department of Neurology, Cerebrovascular Disease Center, Nanjing Brain Hospital Affiliated to Nanjing Medical University, No. 264 Guangzhou Road, Nanjing 210029, China

**Keywords:** E-selectin, brain edema, endovascular therapy, acute ischemic stroke

## Abstract

**Background and Purpose:** Little is known about the effect of soluble adhesion molecules on malignant brain edema (MBE) after endovascular thrombectomy (EVT). This study aimed to explore the association between serum concentrations of E-selectin and the risk of MBE in patients who received EVT. **Methods:** Patients with a large vessel occlusion stroke in the anterior circulation who underwent EVT were prospectively recruited. Serum soluble E-selectin concentrations were measured after admission for all patients. MBE was defined as a midline shift of ≥5 mm on follow-up imaging within 72 h after surgery. Multivariate logistic regression analyses were performed to determine the association between E-selectin levels and the risk of MBE. **Results:** Among the 261 included patients (mean age, 69.7 ± 12.3 years; 166 males), 59 (22.6%) developed MBE. Increasing circulating E-selectin levels were associated with an increased risk of MBE after multivariable adjustment (odds ratios [OR], highest vs. lowest quartile: 3.593; 95% confidence interval [CI], 1.178−10.956; *p* = 0.025). We further observed a significantly positive association between E-selectin and MBE (per 1-standard deviation increase; OR, 1.988; 95% CI, 1.379−2.866, *p* = 0.001) when the E-selectin levels were analyzed as a continuous variable. Furthermore, the restricted cubic spline demonstrated a linear correlation between serum E-selectin levels and the risk of MBE (*p* < 0.001 for linearity). **Conclusions**: In this prospective study, circulating levels of E-selectin were associated with an increased risk of MBE after EVT. Further mechanistic studies are warranted to elucidate the pathophysiology underlying this association.

## 1. Introduction

Stroke is a leading cause of death and disability globally, particularly in low- and middle-income countries, and this burden is increasing [1]. Endovascular thrombectomy (EVT) has been recognized as the standard treatment for large vessel occlusion strokes in the anterior cerebral circulation [2,3,4]. The common complication after EVT is malignant brain edema (MBE), which increases the risk of functional dependence and reduces the beneficial effect of EVT treatment [5,6,7,8]. Hence, determining the predictors of MBE may help in understanding the underlying mechanisms and prompting early identification of patients at risk for brain edema after EVT.

E-selectin, also known as CD62 antigen-like family member E, is an inducible endothelial cell surface molecule of 115 kDa that is specifically expressed on endothelial cells activated by various proinflammatory substances [9]. E-Selectin mediates leukocyte rolling on the endothelium and is thus involved in the recruitment of neutrophils, monocytes, and T cells to inflammatory foci. This process might partly induce endothelial dysfunction and an inflammatory response [10,11]. In the murine stroke model, upregulation of E-selectin expression was observed in the ischemic cerebral vasculature after reperfusion and persisted for 24 h. Blocking E-selectin with a monoclonal antibody increased ischemic cortical cerebral blood flow up to 2.6-fold [12]. Clinical studies indicated that several inflammatory factors, including E-selectin, are significantly higher in stroke patients compared with controls, suggesting a possible predictive role for these markers [13,14]. Furthermore, in patients with ischemic stroke, serum sE-selectin levels were significantly associated with the presence of cerebral microbleeds, which further confirmed that E-selectin is involved in endothelial injury after stroke [15]. However, few studies have investigated the relationship between E-selectin levels and MBE in patients treated with EVT.

We therefore conducted this prospective study to determine whether circulating E-selectin concentrations may predict the presence of MBE in large vessel occlusion stroke patients after EVT treatment.

## 2. Materials and Methods

### 2.1. Study Population

Between September 2019 and July 2021, patients with large vessel occlusion strokes receiving EVT treatment at Nanjing First Hospital were prospectively included. This study included patients who (1) were ≥18 years old; (2) had occlusion of the internal carotid artery or middle cerebral artery confirmed by preoperative imaging; (3) received EVT treatment with a stent-like retriever and/or aspiration system. Patients were excluded if they were diagnosed with a concomitant aneurysm, arteriovenous malformation, moyamoya disease, or leukemia. The local institutional review board approved the study, and all subjects provided informed consent before entering this study.

### 2.2. Baseline Data Collection

Baseline data, including demographic characteristics, clinical data, procedural parameters, and laboratory data, were included in this analysis. Admission neurological deficits were evaluated using the National Institutes of Health Stroke Scale (NIHSS) score [16]. The pre-treatment ischemic change was measured by the Alberta Stroke Program Early Computerized Tomography Score (ASPECTS) [17]. Stroke etiology was classified according to the criteria of the Trial of Org 10172 in Acute Stroke Treatment [18]. The American Society of Interventional and Therapeutic Neuroradiology/Society of Interventional Radiology grading system was utilized to assess the collateral status [19] and was defined as poor (grades 0–1) or good (grades 2–4). Additionally, successful reperfusion was defined as a modified Thrombolysis in Cerebral Infarction score of 2b–3 [20,21]. Whether patients received intravenous thrombolysis before EVT was also recorded in this study.

### 2.3. Blood Sampling and Laboratory Methods

Venous blood was drawn within 24 h of admission. Serum E-selectin levels were measured by the enzyme-linked immunosorbent assay kit (Abcam, Cambridge, UK, Cat#: ab100512). The sensitivity was 30 pg/mL, and the range was 24.69–18,000 pg/mL. The intra- and inter-assay coefficients of variation for E-selectin were <10.0% and <10.0%, respectively. All procedures were performed in strict accordance with the manufacturer’s instructions by a laboratory technician who was blinded to the clinical information.

### 2.4. Diagnosis of MBE

A computed tomography scan was conducted 24–72 h after EVT treatment. According to previous studies [5,22,23], MBE was diagnosed according to the following criteria: (1) parenchymal hypodensity of ≥50% middle cerebral artery territory and signs of local brain swelling such as sulcal effacement and compression of the lateral ventricle; (2) midline shift of ≥5 mm at the septum pellucidum or pineal gland with obliteration of the basal cisterns. The imaging parameters were evaluated by 2 neurologists who were blinded to the clinical data. Disagreements about imaging analysis were resolved through a consensus conference. If disagreements between the reviewers were identified, they were discussed until a consensus was reached.

### 2.5. Statistical Analysis

Categorical variables were expressed as percentages and analyzed using the χ^2^ test or Fisher’s exact test. Continuous variables were demonstrated as mean ± standard deviation or medians (interquartile), analyzed using the Mann–Whitney U test, *t*-test, Kruskal–Wallis test, and one-way analysis of variance, when appropriate. We used 2 multiple-adjusted logistic regression models to explore the relationship between E-selectin levels and MBE. Model 1 was adjusted for age and sex. Model 2 was adjusted for age, sex, and covariates with a *p* value < 0.1 in the univariate analysis, including hypertension, baseline NIHSS score, poor collateral status, successful reperfusion, and baseline blood glucose. Furthermore, restricted cubic splines with 4 knots (at the 5th, 35th, 65th, and 95th percentiles) were conducted to explore the pattern and magnitude of the association between serum E-selectin concentrations and the risk of MBE [24].

A *p* value < 0.05 was considered significant. Statistical analyses were performed with SPSS version 24.0 (IBM, New York, NY, USA) and R statistical software version 4.0.0 (R Foundation, Vienna, Austria).

## 3. Results

### 3.1. Baseline Characteristics of the Study Population

In this study, 283 patients with large artery occlusions receiving EVT treatment were screened for analysis. We excluded 14 patients treated without a stent-like retriever or aspiration system; 5 patients diagnosed with a concomitant aneurysm, arteriovenous malformation, moyamoya disease, or leukemia; and 3 patients without follow-up imaging data for evaluating the MBE. Finally, a total of 261 patients (mean age, 69.7 ± 12.3 years; 166 male) were included in this study. Table 1 demonstrates the baseline data of the study population stratified by the quartiles of serum E-selectin levels. Among these patients, the median time from onset to recanalization was 360.0 min, the median pre-treatment ASPECT score was 9.0, and the baseline NIHSS score was 13.0. Successful recanalization was achieved in 226 (86.6%) patients. Poor collateral circulation status was observed in 143 (54.8%) patients. Additionally, based on the TOAST criteria, 46.0% of patients were diagnosed with large-artery atherosclerosis, and 43.7% of patients were diagnosed with cardio-embolism.

### 3.2. Incidence and Factors Associated with MBE

During hospitalization, 59 (22.6%) developed MBE. The clinical characteristics comparing patients with and without MBE are shown in Table 2. On univariate analysis, hypertension was more common in patients with MBE than in patients without (83.1% versus 68.3%; *p* = 0.027). Patients with MBE had a higher baseline NIHSS score (median, 12.0 versus 12.0; *p* = 0.001), blood glucose levels (mean, 8.0 versus 6.9 mmol/L, *p* = 0.002), and E-selectin levels (median, 8.1 versus 6.8 ng/mL, *p* = 0.001; Figure 1). Furthermore, poor collateral circulation status (79.7% versus 47.5%; *p* = 0.001) and unsuccessful recanalization (35.6% versus 6.9%; *p* = 0.001) were higher in patients with MBE than in patients without MBE.

### 3.3. Correlations between E-Selectin Levels and MBE

In multivariate analysis after adjusting for potential confounders, E-selectin levels were associated with an increased risk of MBE after multivariable adjustment (odds ratios [OR], highest vs. lowest quartile: 3.593; 95% confidence interval [CI], 1.178−10.956; *p* = 0.025) (Table 3). We further observed a significantly positive association between E-selectin and MBE (per 1-standard deviation increase; OR, 1.988; 95% CI, 1.379−2.866, *p* = 0.001) when the E-selectin levels were analyzed as a continuous variable. Furthermore, the restricted cubic spline demonstrated a linear correlation between serum E-selectin levels and the risk of MBE (*p* < 0.001 for linearity; Figure 2).

## 4. Discussion

In this prospective study, higher serum levels of E-selectin were statistically significantly associated with an increased risk of MBE in ischemic stroke patients receiving EVT treatment. This association remained significant after adjusting for confounders including age, sex, hypertension, baseline NIHSS score, poor collateral status, successful reperfusion, and baseline blood glucose.

As one of the serious complications in acute ischemic stroke patients after EVT treatment, MBE might lead to a higher rate of functional dependence mortality at 90 days [5,6,7,8]. Davoli et al. analyzed patients from a prospective single-center database and found that in patients with large intracranial vessel occlusions undergoing EVT, 35.7% of patients developed malignant middle cerebral artery infarction [25]. Then, another study demonstrated that 43.7% of patients experienced midline shifts after EVT [6]. However, our prospective study showed that 22.6% of patients developed MBE, which is similar to data from a retrospective analysis of 130 patients (25.6% of patients with MBE) [5]. Furthermore, previous clinical studies have found several predictive factors for the presence of MBE in patients following EVT [5,6,25,26]. These factors included advanced age, fasting blood glucose, hypertension, baseline NIHSS score, ASPECT score, collateral score, and unsuccessful recanalization. However, this study did not find a significant association between MBE and age or hypertension. Previous researchers investigating the MBE used inconsistent definitions, leading to discrepancies in incidence rates and associated factors. These data indicated the need for an internationally agreed-upon definition of MBE.

E-selectin is a glycoprotein adhesion molecule specifically expressed on activated endothelial cells after ischemic stimulation [27]. As described previously, E-selectin is an independent predictor of clinical outcomes in a population of patients with cerebrovascular diseases [28]. Our study extended the current knowledge about the adverse effect of E-selectin in ischemic stroke as it demonstrated a negative association between E-selectin concentrations and MBE in ischemic stroke patients after EVT. The mechanisms by which circulating E-selectin affects MBE after mechanical recanalization are unclear, but several potential pathophysiological pathways have been suggested. First, increased E-selectin might indicate dysfunction of the vascular endothelium, leading to a subsequent breakdown of blood–brain barrier function after reperfusion [29,30]. Second, as described previously, immunomodulatory and anti-inflammatory cytokine release is targeted to activate blood vessels by the E-selectin antigen specificity of regulatory T cells, which are believed to be associated with brain tissue damage after stroke [29,30,31]. Thirdly, it has been reported that vascular endothelium expression of E-selectin occurs in the first few minutes to hours after ischemic stroke [12,32,33]. This process might induce leukocyte or neutrophil migration into brain tissue, which further contributes to cytokine release and free radical-mediated damage. Finally, hyperglycemia is likely to represent an important link between these two conditions. Our results demonstrated that increased E-selectin levels were associated with baseline blood glucose, and blood glucose has been confirmed as a significant predictor of MBE [5]. Further studies are needed to clarify this potential mechanism.

This study should be approached with caution due to its limitations. First, the E-selectin levels used in this study were only measured from one blood test after admission. Determining the longitudinal changes in circulating E-selectin levels could provide meaningful insights into the presence of MBE. Second, although we utilized a multivariate model to predict MBE with a high degree of predictive accuracy, there may be other confounders and unmeasured markers of MBE. Finally, this was a single-center, small-scale study, which might lead to a selective bias. Therefore, our results need to be confirmed in an external cohort before they can be generalized to different populations. Also, experimental studies are warranted to elucidate the pathophysiologic mechanisms that may underlie the described relationship between E-selectin and MBE risk.

In summary, our data demonstrated that elevated circulating E-selectin levels were significantly associated with the risk of MBE after EVT. Whether treatment of blood–brain barrier dysfunction in patients with large vessel occlusion would improve outcomes is uncertain, but measurement of serum E-selectin may serve as a predictor for the risk of blood–brain barrier breakdown. Further investigations to determine whether E-selectin is a modifiable target to reduce MBE are needed.

## Figures and Tables

**Figure 1 brainsci-13-01097-f001:**
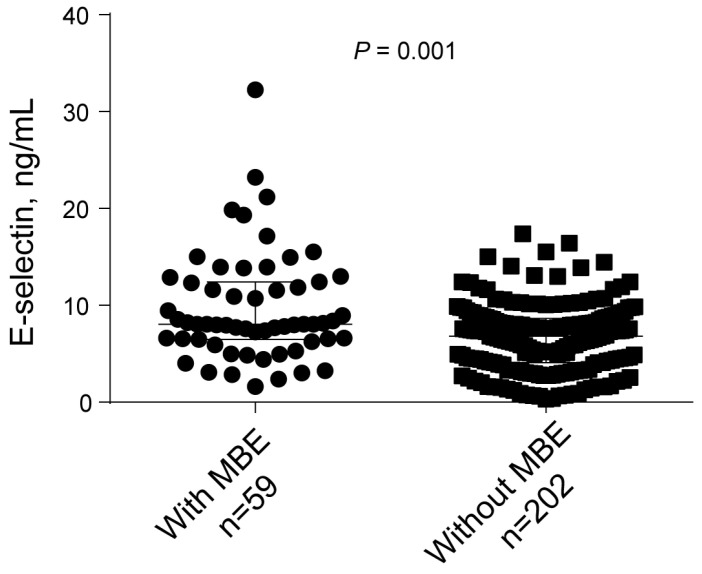
Univariate analysis of the E-selectin levels with MBE.

**Figure 2 brainsci-13-01097-f002:**
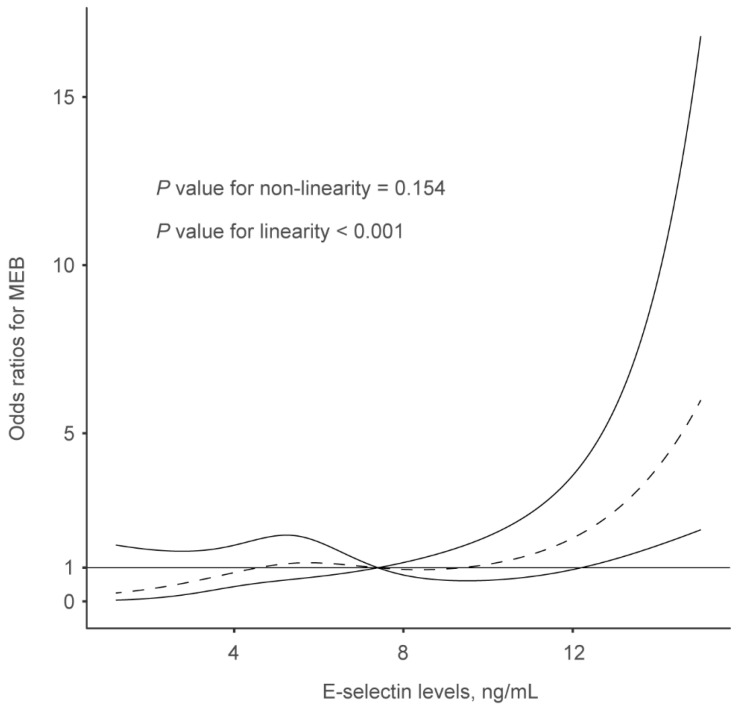
Restricted cubic spline was utilized to explore the relationship between serum E-selectin levels and malignant brain edema, which fitted with four knots (at the 5th, 35th, 65th, and 95th percentiles), adjusting for covariates with *p* values < 0.1 in the univariate analysis, including hypertension, baseline NIHSS score, poor collateral status, successful reperfusion, and baseline blood glucose. The median serum E-selectin levels are set as the reference point.

**Table 1 brainsci-13-01097-t001:** Baseline data of the study population stratified by the quartile of E-selectin levels.

Variables	All Patients, *n* = 261	E-Selectin Quartiles				*p* Value
		First, *n* = 67	Second, *n* = 65	Third, *n* = 65	Fourth, *n* = 64	
Demographic characteristics						
Age, years	69.7 ± 12.3	67.8 ± 12.9	70.7 ± 9.7	70.6 ± 13.6	69.8 ± 12.6	0.494
Male, *n* (%)	166 (63.6)	40 (59.7)	40 (61.5)	42 (64.4)	44 (68.8)	0.724
Medical history, *n* (%)						
Hypertension	187 (71.6)	51 (76.1)	43 (66.2)	43 (66.2)	50 (78.1)	0.271
Diabetes mellitus	66 (25.3)	13 (19.4)	16 (24.6)	20 (30.8)	17 (26.6)	0.506
Hyperlipidemia	28 (10.7)	3 (4.5)	8 (12.3)	7 (10.8)	10 (15.6)	0.212
Current smoker	103 (39.5)	28 (41.8)	24 (36.9)	27 (41.5)	24 (37.5)	0.908
Alcohol intake	69 (26.5)	20 (29.9)	18 (27.7)	17 (26.6)	14 (21.9)	0.768
Coronary heart disease	36 (13.8)	6 (9.0)	11 (16.9)	12 (18.5)	7 (10.9)	0.323
Clinical data						
Systolic blood pressure, mmHg	136.4 ± 22.6	139.3 ± 23.4	134.6 ± 19.1	138.0 ± 23.8	133.6 ± 23.9	0.416
Diastolic blood pressure, mmHg	82.7 ± 13.5	85.0 ± 11.9	80.1 ± 14.0	83.9 ± 14.3	81.7 ± 13.5	0.147
Time from onset to blood sample, h	19.0 (13.0, 23.5)	18.0 (11.0, 24.0)	16.0 (13.0, 23.0)	21.0 (16.0, 24.0)	19.0 (13.0, 23.0)	0.136
Time from onset to recanalization, min	360.0 (248.0, 569.0)	400.0 (255.0, 630.0)	330.5 (231.0, 423.0)	360.0 (243.0, 665.0)	366.0 (275.0, 555.0)	0.171
Admission NIHSS, score	13.0 (10.0, 17.0)	12.0 (8.0, 15.0)	13.0 (10.0, 18.0)	14.0 (12.0, 17.0)	15.0 (10.0, 19.0)	0.015
Pre-treatment ASPECTS, score	9.0 (8.0, 9.0)	9.0 (8.0, 9.0)	9.0 (8.0, 9.0)	8.0 (8.0, 9.0)	8.0 (8.0, 9.0)	0.628
Malignant brain edema, *n* (%)	59 (22.6)	8 (11.9)	13 (20.0)	16 (24.6)	22 (34.4)	0.020
Stroke etiology, *n* (%)						0.902
Atherosclerotic	120 (46.0)	26 (38.8)	30 (46.2)	33 (50.8)	31 (48.4)	
Cardioembolic	114 (43.7)	33 (49.3)	28 (43.1)	26 (40.0)	27 (42.2)	
Undetermined or others	27 (10.3)	8 (11.9)	7 (10.8)	6 (9.2)	6 (9.4)	
Use of intravenous alteplase, *n* (%)	120 (46.0)	28 (41.8)	32 (49.2)	28 (43.1)	32 (50.0)	0.709
Poor collateral circulation, *n* (%)	143 (54.8)	36 (53.7)	33 (50.8)	34 (52.3)	40 (62.5)	0.541
Successful recanalization, *n* (%)	226 (86.6)	59 (88.1)	56 (86.2)	52 (80.0)	59 (92.2)	0.231
Occlusive vessel, *n* (%)						0.994
Internal carotid artery	83 (31.8)	22 (32.8)	20 (30.8)	21 (32.3)	20 (31.3)	
Middle cerebral artery	178 (68.2)	45 (67.2)	45 (69.2)	44 (67.7)	44 (68.8)	
90-day mRS, score	3.0 (1.0, 5.0)	2.0 (0, 4.0)	3.0 (1.0, 4.5)	3.0 (1.0, 4.0)	3.5 (2.0, 6.0)	0.034
Laboratory data						
Baseline blood glucose, mmol/L	7.2 ± 2.4	6.4 ± 1.7	7.1 ± 2.5	7.3 ± 2.6	8.1 ± 2.5	0.001
Hs-CRP, mg/L	10.9 (4.4, 23.9)	8.7 (3.7, 19.3)	10.7 (4.2, 28.3)	9.3 (3.5, 20.5)	20.0 (5.3, 38.5)	0.068

Abbreviations: ASPECTS, the Alberta Stroke Program Early Computed Tomography Score; Hs-CRP, hypersensitive C-reactive protein; NIHSS, the National Institute of Health Stroke Scale.

**Table 2 brainsci-13-01097-t002:** Comparison of the baseline data in patients with and without malignant brain edema.

Variables	Malignant Brain Edema		*p* Value
	Yes, *n* = 59	No, *n* = 202	
Demographic characteristics			
Age, years	71.1 ± 13.0	69.2 ± 12.1	0.304
Male, *n* (%)	39 (66.1)	127 (62.9)	0.650
Medical history, *n* (%)			
Hypertension	49 (83.1)	138 (68.3)	0.027
Diabetes mellitus	15 (25.4)	51 (25.2)	0.978
Hyperlipidemia	4 (6.8)	24 (11.9)	0.265
Current smoker	25 (42.4)	78 (38.6)	0.603
Alcohol intake	11 (18.6)	58 (28.9)	0.128
Coronary heart disease	11 (18.6)	25 (12.4)	0.219
Clinical data			
Systolic blood pressure, mmHg	135.6 ± 22.8	136.7 ± 22.7	0.754
Diastolic blood pressure, mmHg	82.8 ± 13.8	82.6 ± 13.4	0.926
Time from onset to blood sample, h	18.0 (13.0, 22.5)	19.0 (13.0, 24.0)	0.514
Time from onset to recanalization, min	360.0 (236.0, 580.0)	360.0 (250.0, 567.0)	0.859
Admission NIHSS, score	16.0 (12.0, 20.0)	12.0 (9.0, 16.0)	0.001
Pre-treatment ASPECTS, score	8.0 (8.0, 9.0)	9.0 (8.0, 9.0)	0.294
Stroke etiology, *n* (%)			0.364
Atherosclerotic	25 (42.4)	95 (47.0)	
Cardioembolic	25 (42.4)	89 (44.1)	
Undetermined or others	9 (15.3)	18 (8.9)	
Use of intravenous alteplase, *n* (%)	30 (50.8)	90 (44.6)	0.394
Poor collateral circulation, *n* (%)	47 (79.7)	96 (47.5)	0.001
Successful recanalization, *n* (%)	38 (64.4)	188 (93.1)	0.001
90-day mRS, score	4.0 (3.0, 6.0)	2.0 (1.0, 4.0)	0.001
Occlusive vessel, *n* (%)			0.178
Internal carotid artery	23 (39.0)	60 (29.7)	
Middle cerebral artery	36 (61.0)	142 (70.3)	
Laboratory data			
Baseline blood glucose, mmol/L	8.0 ± 2.9	6.9 ± 2.2	0.002
Hs-CRP, mg/L	11.0 (4.6, 30.5)	10.8 (4.1, 22.0)	0.181
E-selectin, ng/mL	8.1 (6.5, 12.4)	6.8 (4.1, 8.6)	0.001

Abbreviations: ASPECTS, the Alberta Stroke Program Early Computed Tomography Score; Hs-CRP, hypersensitive C-reactive protein; NIHSS, the National Institute of Health Stroke Scale.

**Table 3 brainsci-13-01097-t003:** Binary regression analysis for the associations of serum E-selectin levels with malignant brain edema.

Variables	Crude Model		Model 1		Model 2	
	OR (95% CI)	*p* Value	OR (95% CI)	*p* Value	OR (95% CI)	*p* Value
E-selectin levels (per 1-SD increase)	1.830 (1.357−2.468)	0.001	1.834 (1.356−2.480)	0.001	1.988 (1.379−2.866)	0.001
E-selectin levels (quartiles)						
First	Reference		Reference		Reference	
Second	1.844 (0.709−4.798)	0.210	1.780 (0.682−4.646)	0.239	1.712 (0.539−5.436)	0.362
Third	2.408 (0.951−6.100)	0.064	2.311 (0.908−5.881)	0.079	1.974 (0.635−6.140)	0.240
Fourth	3.863 (1.569−9.509)	0.003	3.731 (1.509−9.226)	0.004	3.593 (1.178−10.956)	0.025

Abbreviations: CI, confidence interval; OR, odds ratio; SD, standard deviation. Model 1 adjusted age and sex. Model 2 adjusted for demographic characteristics and variables with a *p* value < 0.1 in the univariate analysis, including hypertension, baseline NIHSS score, poor collateral status, successful reperfusion, and baseline blood glucose.

## Data Availability

Data are available from the corresponding author upon reasonable request.

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
