# Peer review of "Higher Serum E-Selectin Levels Associated with Malignant Brain Edema after Endovascular Thrombectomy for Ischemic Stroke: A Pilot Study"

_brainsci, 2023, doi:10.3390/brainsci13071097_

Round 1
Reviewer 1 Report
The article is well designed and the topic is interesting. However, I consider that there are several points for improvement and elements that are not well explained.
- In the introduction, the authors comment that E-selectin mediates endothelial dysfunction but do not explain how. I think they should add information about it.
- In the "study population" subsection, they state that patients with "hematological system diseases" have been excluded. What are those diseases?
- It is absolutely necessary to add a flow chart that explains the total number of patients admitted with acute ischemic stroke and the different exclusions. I don't know if it's in the unpublished material as it won't let me open it. Anyway I think it should be in the material to publish.
- The following times should be indicated in detail: LTSW to MT, LTSW to blood sample, initial CT to recanalization, CT to sample, recanalization to sample.
- In the "baseline data" section, they should correct referring to ASPECTS as a method to determine pre-infarction volume. In any case, they should measure said volume using CBF in CT-perfusion or DWI in MRI depending on the technique of choice.
- In general, enough details are missing that would help to better understand the study population and the blood findings. The final TICI is not indicated in all the patients or in the groups (only successful recanalization), the time to the control CT, the presence of tandem occlusions, the Rankin scale at discharge and at 90 days, etc.
With all that said, I believe that quite a few corrections have to be made before publishing the paper.
Throughout the text I have identified several lexical and grammatical errors that must be corrected. For example, "occlusive" has to be changed to occlusion in "large vessel occlusion"; just like "firstly" or "secondly" for "first" and "second". I consider that all the text must be reviewed by a native English speaker.
Author Response
Reviewer 1
- The article is well designed and the topic is interesting. However, I consider that there are several points for improvement and elements that are not well explained.
Reply: Thank you for your careful reading and positive comment.
- In the introduction, the authors comment that E-selectin mediates endothelial dysfunction but do not explain how. I think they should add information about it.
Reply: Thanks for this helpful advice. We now added several information about E-selectin and endothelial dysfunction in the Introduction (Line 47- Line 52).
- In the "study population" subsection, they state that patients with "hematological system diseases" have been excluded. What are those diseases?
Reply: Thanks for your comment. Patients with a history of leukemia were excluded from this study. We now modified the description in the revised paper.
- It is absolutely necessary to add a flow chart that explains the total number of patients admitted with acute ischemic stroke and the different exclusions. I don't know if it's in the unpublished material as it won't let me open it. Anyway I think it should be in the material to publish.
Reply: Thanks for your helpful advice. In this study, 283 patients with large artery occlusion receiving EVT treatment were screened for analysis. We excluded 14 patients treated without stent-like retriever or aspiration system, 5 patients diagnosed with a concomitant aneurysm, arteriovenous malformation, moyamoya disease, or leukemia, and 3 patients without follow-up im-aging data for evaluating the MBE. Finally, a total of 261 patients (mean age, 69.7 ± 12.3 years; 166 male) were included in this study. We now added this important data in the revised paper (Line 122-Line 127).
- The following times should be indicated in detail: LTSW to MT, LTSW to blood sample, initial CT to recanalization, CT to sample, recanalization to sample.
Reply: Thanks for your helpful advice. The time from onset to blood sample and time from onset to recanalization were added and analyzed in the revised paper.
- In the "baseline data" section, they should correct referring to ASPECTS as a method to determine pre-infarction volume. In any case, they should measure said volume using CBF in CT-perfusion or DWI in MRI depending on the technique of choice.
Reply: Thanks for your helpful advice. The authors understand your concern. The ASPECTS is considered as a systematic method of assessing ischemic change on non-contrast CT scans (Line 80-Line 81). We now modified the description in the revised manuscript.
- In general, enough details are missing that would help to better understand the study population and the blood findings. The final TICI is not indicated in all the patients or in the groups (only successful recanalization), the time to the control CT, the presence of tandem occlusions, the Rankin scale at discharge and at 90 days, etc. With all that said, I believe that quite a few corrections have to be made before publishing the paper.
Reply: Thanks for your helpful advice. This study defined the mTICI score of 2b–3 as successful reperfusion. We therefore did not report the final TICI in the manuscript. Furthermore, we added the 90-day mRS score in the revised paper following your suggestion.
- Throughout the text I have identified several lexical and grammatical errors that must be corrected. For example, "occlusive" has to be changed to occlusion in "large vessel occlusion"; just like "firstly" or "secondly" for "first" and "second". I consider that all the text must be reviewed by a native English speaker.
Reply: Thanks for your careful reading. We now modified the lexical and grammatical errors. Also, the revised manuscript was reviewed by a native English speaker.
Reviewer 2 Report
In this study, increasing circulating E-selectin levels were associated with an increased risk of MBE after multivariable adjustment.
The authors conducted the study appropriately in general, but it has some deficiencies that weaken the strength of the work.
Firstly, the authors solely focus on E-selectin, which plays a crucial role in the recruitment and adhesion of leukocytes to the endothelium during inflammation and immune responses, as well as other processes such as tissue remodeling, immune cell activation, and serving as an inflammation mediator. However, this reviewer considers that the introduction is somewhat limited in supporting the highlighted effects, and recommends a slight expansion within the permitted length limits.
The presented work is a prospective study, and it draws attention that the authors did not include a more comprehensive study to clarify the role of E-selectin. It would be advisable to perform analyses at various time points due to the nature of the inflammatory process associated with edema and mostly time-dependent. Additionally, it would be recommended to incorporate the analysis of other molecules from different inflammatory pathways or related to endothelial stability that could provide a more effective understanding of the role of E-selectin.
The current study only allows for a simple association of a molecule that has already been extensively studied in the field of cerebral ischemia. Furthermore, as stated by the authors, it is a small and single-center study limiting the robustness of the results. This reviewer would appreciate a more in-depth discussion regarding the possible relationship between E-selectin and MBE, as the discussion currently only provides an expansion of the results without profound interpretations from expert personnel such as the authors.
Additionally, some minor suggestions:
Line 39: It is recommended to use global data for a more comprehensive understanding of the studied process by the readers.
Author Response
In this study, increasing circulating E-selectin levels were associated with an increased risk of MBE after multivariable adjustment. The authors conducted the study appropriately in general, but it has some deficiencies that weaken the strength of the work.
Reply: Thank you for your careful reading and comment.
- Firstly, the authors solely focus on E-selectin, which plays a crucial role in the recruitment and adhesion of leukocytes to the endothelium during inflammation and immune responses, as well as other processes such as tissue remodeling, immune cell activation, and serving as an inflammation mediator. However, this reviewer considers that the introduction is somewhat limited in supporting the highlighted effects, and recommends a slight expansion within the permitted length limits.
Reply: Thanks for this helpful comment. A detailed description of E-selectin in mediating the recruitment and adhesion of leukocytes to the endothelium was added in the Introduction (Line 47- Line 52).
- The presented work is a prospective study, and it draws attention that the authors did not include a more comprehensive study to clarify the role of E-selectin. It would be advisable to perform analyses at various time points due to the nature of the inflammatory process associated with edema and mostly time-dependent. Additionally, it would be recommended to incorporate the analysis of other molecules from different inflammatory pathways or related to endothelial stability that could provide a more effective understanding of the role of E-selectin.
Reply: Thanks for this comment. E-selectin levels were only measured from one blood test after admission. Determining the longitudinal changes of circulating E-selectin levels could provide meaningful insights into the presence of MBE. We now discussed this information as a limitation in the revised manuscript (Line 251-Line 218). In addition, this study also assessed the predictive value of Hs-CRP. However, we did not find a significant correlation between Hs-CRP levels and the risk of MBE. In future studies, we will evaluate the role of brain injury markers and inflammatory cytokines in predicting MBE.
- The current study only allows for a simple association of a molecule that has already been extensively studied in the field of cerebral ischemia. Furthermore, as stated by the authors, it is a small and single-center study limiting the robustness of the results. This reviewer would appreciate a more in-depth discussion regarding the possible relationship between E-selectin and MBE, as the discussion currently only provides an expansion of the results without profound interpretations from expert personnel such as the authors.
Reply: Thank you for noting this issue. A detailed description of E-selectin in mediating MBE was added in the Discussion (Line 204- Line 213).
- Line 39: It is recommended to use global data for a more comprehensive understanding of the studied process by the readers.
Reply: Thanks for this comment. The epidemiological data of stroke in worldwide was added in the revised paper.
Reviewer 3 Report
Here are the comments on the manuscript:
1. the English language must be checked by a native speaker,
2. the role of E-selectin as a predictor of stroke and its complications is well known - so what's so special about this study? The authors need to make this more clear.
3. were patients with COVID-19 included in the study?
4. more clinical data is needed here. What stents were used? Did the same clinician always perform the thrombectomy?
5. patients also underwent intravenous thrombolysis, and the authors do not mention it in the methodology. Please explain.
6. Other outcomes are known or obvious to predict. Therefore, the work does not contribute much to understanding the etiopathogenesis of ischemic stroke.
The English language must be checked by a native speaker.
Author Response
- the English language must be checked by a native speaker,
Reply: Thanks for your careful reading. The revised manuscript was reviewed by a native English speaker.
- the role of E-selectin as a predictor of stroke and its complications is well known - so what's so special about this study? The authors need to make this more clear.
Reply: Thanks for your comment. A previous study has confirmed that E-selectin is an independent predictor of clinical outcomes in a population of patients with cerebrovascular diseases (J Inflamm. 2015; 12:61). However, whether serum E-selectin levels associated with stroke complications, such as hemorrhagic transformation, early neurological deterioration, and MBE are still unknown. Therefore, we conducted this prospective study to determine whether circulating E-selectin concentrations may predict the presence of MBE in large vessel occlusion stroke patients after EVT treatment.
- were patients with COVID-19 included in the study?
Reply: Thank you for your comment. This study sample did not include patients with COVID-19.
- more clinical data is needed here. What stents were used? Did the same clinician always perform the thrombectomy?
Reply: Thank you for your comment. In this study, only a small number of patients received stent implantation during EVT treatment. We therefore did not report this data. In addition, the EVT was performed by a stroke team, which included 5 neurointerventionists.
- patients also underwent intravenous thrombolysis, and the authors do not mention it in the methodology. Please explain.
Reply: Thank you for this comment. Whether patients received intravenous thrombolysis before EVT was also recorded in this study. We now added this important information in the methodology (Line 86-Line 87).
- Other outcomes are known or obvious to predict. Therefore, the work does not contribute much to understanding the etiopathogenesis of ischemic stroke.
Reply: Authors understand your concerns. As described previously, E-selectin is an independent predictor of clinical outcomes in a population of patients with cerebrovascular diseases (J Inflamm. 2015; 12:61). Our study extended the current knowledge about the adverse effect of E-selectin in ischemic stroke as it demonstrated a negative association between E-selectin concentrations and MBE in ischemic stroke patients after EVT. To the best of our knowledge, this is the first study to detect the value of E-selectin in predicting the presence of MBE. Our data demonstrated that circulating levels of E-selectin were associated with an increased risk of MBE after EVT. Further mechanistic studies are warranted to elucidate the pathophysiology underlying this association.
Reviewer 4 Report
we read with interest the article by Zhou et al assessing levels of serum E-selectin with malignant brain edema after endovascular thrombectomy for ischemic stroke. The authors have found circulating levels of E-selectin were associated with an increased risk of MBE after EVT.
This work has a number of limitations partly described by the authors; however, these limitations are major in the findings which render this work a pilot study rather being a full-pledged study.
First for the methods:
The authors should describe their experimental procedure as when the blood was taken and what is the full description of the ELISA kit used in terms of sensitivity, LOD, LLOD, and specificity; these are totally missing and the description of quartiles is confusing as one would think the PIs are collecting 4 blood testing at different time points which is not the case. Please elaborate in the method section.
Results:
please plot the data of the E-Selectin as a dot-plot for each patient in each group as it is important to the readers to show the trend of E-selectin levels.
the work should include a control cohort that is lacking to set the baseline and remove the effect of surgery on E-Selectin. This is a major weakness in the study
Discussion
the work should be including other comorbid conditions such as weight, smoking, and alcohol intake as these can affect the glycosylation of the glycoprotein E selectin and this was missing,
another point to consider is that this protein is a cell surface protein. the authors should have highlighted one mechanism of how this protein is sieving into the blood as it' is expressed in cells.
Other markers should be assessed such as brain injury markers as well as inflammatory cytokines
Conclusion
the work is a pilot study as it has major limitations in design nd the experimental execution of this work.
Author Response
we read with interest the article by Zhou et al assessing levels of serum E-selectin with malignant brain edema after endovascular thrombectomy for ischemic stroke. The authors have found circulating levels of E-selectin were associated with an increased risk of MBE after EVT. This work has a number of limitations partly described by the authors; however, these limitations are major in the findings which render this work a pilot study rather being a full-pledged study.
Reply: Thank you for your careful reading and comment.
- The authors should describe their experimental procedure as when the blood was taken and what is the full description of the ELISA kit used in terms of sensitivity, LOD, LLOD, and specificity; these are totally missing and the description of quartiles is confusing as one would think the PIs are collecting 4 blood testing at different time points which is not the case. Please elaborate in the method section.
Reply: Thank you for your helpful comment. Venous blood was drawn within 24 hours of admission. Serum E-selectin levels were measured by the enzyme-linked immunosorbent assay kit (Abcam UK, Cat#: ab100512). The sensitivity was 30 pg/ml, and the range was 24.69 pg/ml - 18000 pg/ml. The intra- and inter-assay coefficients of variation for E-selectin were <10.0% and <10.0%, respectively. All procedures were performed in strict accordance with the manufacturer’s instructions by a laboratory technician who was blinded to the clinical information. A detailed description of the measurement of E-selectin levels was added in the manuscript (Line 89-Line 94).
- please plot the data of the E-Selectin as a dot-plot for each patient in each group as it is important to the readers to show the trend of E-selectin levels.
Reply: Thanks for this comment. The dot-plot for each patient in each group was added in Figure 1 in the revised paper.
- the work should include a control cohort that is lacking to set the baseline and remove the effect of surgery on E-Selectin. This is a major weakness in the study
Reply: Authors understand your concerns. Although this study did not evaluate the E-selectin levels in the healthy control group, the restricted cubic spline confirmed a linear correlation between serum E-selectin levels and the risk of MBE (P < 0.001 for linearity), which indicated that the E-selectin might involve in the pathophysiology of MBE.
- the work should be including other comorbid conditions such as weight, smoking, and alcohol intake as these can affect the glycosylation of the glycoprotein E selectin and this was missing.
Reply: Thanks for this helpful advice. The smoking and alcohol intake were added and analyzed in the revised manuscript.
- another point to consider is that this protein is a cell surface protein. the authors should have highlighted one mechanism of how this protein is sieving into the blood as it' is expressed in cells.
Reply: Thanks for this comment. After endothelial cell stimulation, newly synthesized E-selectin is rapidly detected. This rapid upregulation, although not completely understood, has been explained by the release of a soluble form of E-selectin.
- Other markers should be assessed such as brain injury markers as well as inflammatory cytokines
Reply: Thank you for your helpful suggestion. In this study, we also assessed the predictive value of Hs-CRP. However, we did not find a significant correlation between Hs-CRP levels and the risk of MBE. In future studies, we will evaluate the role of brain injury markers and inflammatory cytokines in predicting MBE.
Round 2
Reviewer 2 Report
The authors have responded adequately to the modifications in the article itself.
Author Response
The authors have responded adequately to the modifications in the article itself.
Reply: Authors thanks for your help.
Reviewer 3 Report
The authors have addressed all of my concerns with the original manuscript. The revised manuscript is ready for publication.
Author Response
The authors have addressed all of my concerns with the original manuscript. The revised manuscript is ready for publication.
Reply: Authors thanks for your help.
Reviewer 4 Report
we thank the reviewers for their efforts, still, the study has limitations that affect the generalizability of the results:
first th, it is a small and single-center study limiting the robustness of the results, second the control cohorts are absent.
I would accept the work if the study is labeled as a Pilot study:
with the title:
Higher serum E-selectin levels were associated with malignant brain edema after endovascular thrombectomy for ischemic stroke: A pilot study
Author Response
I would accept the work if the study is labeled as a Pilot study: with the title: Higher serum E-selectin levels were associated with malignant brain edema after endovascular thrombectomy for ischemic stroke: A pilot study
Reply: Thanks for your comment. The title was changed following your suggestion in the revised manuscript.